# Profiling the Oral Microbiome and Plasma Biochemistry of Obese Hyperglycemic Subjects in Qatar

**DOI:** 10.3390/microorganisms7120645

**Published:** 2019-12-03

**Authors:** Muhammad U. Sohail, Mohamed A. Elrayess, Asma A. Al Thani, Maha Al-Asmakh, Hadi M. Yassine

**Affiliations:** 1Biomedical Research Center, Qatar University, Doha 2713, Qatar; m.elrayess@qu.edu.qa (M.A.E.); aaja@qu.edu.qa (A.A.A.T.); maha.alasmakh@qu.edu.qa (M.A.-A.); 2Department of Biomedical Sciences, College of Health Sciences, QU Health, Qatar University, Doha 2713, Qatar

**Keywords:** obesity, diabetes, pre-diabetes, Qatar Biobank, oral microbiome, testosterone, lipid profile

## Abstract

The present study is designed to compare demographic characteristics, plasma biochemistry, and the oral microbiome in obese (*N* = 37) and lean control (*N* = 36) subjects enrolled at Qatar Biobank, Qatar. Plasma hormones, enzymes, and lipid profiles were analyzed at Hamad Medical Cooperation Diagnostic Laboratory. Saliva microbiome characterization was carried out by 16S rRNA amplicon sequencing using Illumina MiSeq platform. Obese subjects had higher testosterone and sex hormone-binding globulin (SHBG) concentrations compared to the control group. A negative association between BMI and testosterone (*p* < 0.001, r = −0.64) and SHBG (*p* < 0.001, r = −0.34) was observed. Irrespective of the study groups, the oral microbiome was predominantly occupied by *Streptococcus*, *Prevotella*, and *Veillonella* species. A generalized linear model revealed that the Firmicutes/Bacteroidetes ratio (2.25 ± 1.83 vs. 1.76 ± 0.58; corrected *p*-value = 0.04) was higher, and phylum Fusobacteria concentration (4.5 ± 3.0 vs. 6.2 ± 4.3; corrected *p*-value = 0.05) was low in the obese group compared with the control group. However, no differences in microbiome diversity were observed between the two groups as evaluated by alpha (Kruskal–Wallis *p* ≥ 0.78) and beta (PERMANOVA *p* = 0.37) diversity indexes. Certain bacterial phyla (Acidobacteria, Bacteroidetes, Fusobacteria, Proteobacteria, Spirochaetes, and Firmicutes/Bacteroidetes) were positively associated (*p* = 0.05, r ≤ +0.5) with estradiol, fast food consumption, creatinine, breastfed during infancy, triglycerides, and thyroid-stimulating hormone concentrations. In conclusion, no differences in oral microbiome diversity were observed between the studied groups. However, the Firmicutes/Bacteroidetes ratio, a recognized obesogenic microbiome trait, was higher in the obese subjects. Further studies are warranted to confirm these findings in a larger cohort.

## 1. Introduction

Diabetes mellitus is a chronic metabolic disorder with devastating consequences. Genetics and environment are the two main etiological factors that partially or collectively contribute to type 2 diabetes mellitus (T2DM) pathology. In most cases, several genes and epigenetic factors are involved in the development of the disease [1]. However, the significant rise in T2DM incidences suggests that environmental factors play a relatively more important role in the disease pathology than we previously thought. Many environmental factors, such as diet and physical activity, play a crucial role in T2DM development and pathogenesis [2]. Accordingly, a better understanding of the disease pathology and etiology could improve prevention and treatment options. However, no single-time, definitive therapy is available for the treatment of diabetes, and mostly the disease is managed by continuous, long-term medications. In this regard, exploring potential preventative and therapeutic measures requires a better understanding of the long pre-diabetic phase that can last for several years [3]. Hyperglycemia in the pre-diabetic stage is correlated with obesity, blood pressure, and dyslipidemia [4]. However, like diabetes, obesity also does not have a well-defined etiology or pathogenesis and, in many cases, obesity is considered as a prodrome of insulin resistance (IR) in diabetes [5].

Evaluation and characterization of the mucosal microbiome as an environmental factor for obesity and diabetes has received much attraction in recent years. Several studies showed that alterations in the gut microbiome could be liked with IR and obesity and that the microbiome could be a hopeful therapeutic goal for prevention of these metabolic syndromes [6]. Furthermore, microbiome studies in diabetic patients revealed a strong correlation between microbiome diversity and obese/diabetic phenotypes [7,8]. Particularly, elevated Firmicutes and Proteobacteria populations and a drop in Bacteroidetes levels are reported in obese and IR subjects [9,10]. However, most of the published literature is focused on patients with established diabetes, and very few studies have explored the microbiome of the pre-diabetic phase of hyperglycemia [11,12]. Although the oral microbiome has been shown to recapitulate the gut microbiome [13], it is relatively less explored. Accordingly, the association between the oral microbiome and disease prognosis is not well defined.

The oral microbiome has been implicated in several oral cavity (dental caries, periodontitis, endodontic, alveolar osteitis, and tonsillitis) and systemic diseases (cardiovascular disease, stroke, pneumonia, and diabetes) [14,15,16,17]. Long et al. [18] reported that several bacterial taxa in the phylum Actinobacteria are negatively associated with obesity and diabetes. Similarly, many other studies have also reported phylogenetic differences in the microbiome profile of obese and lean humans [19,20]. However, the compositional changes in microbial ecology that are observed in diabetic and obese subjects are often not in agreement across these studies. This makes it challenging to interpret the role of the microbiome in disease development and pathogenicity. We, therefore, sought to characterize the oral microbial community as a biomarker of obesity and diabetes. To achieve this objective, we studied the oral microbiome in obese hyperglycemic subjects. 

## 2. Materials and Methods

### 2.1. Study Cohort

This study enrolled obese pre-diabetic (hyperglycemic) adult males and age-matched healthy individuals who voluntarily enrolled at Qatar Biobank (QBB; https://www.qatarbiobank.org.qa/). The inclusion criteria for the participants were: age ≥ 30 years, body mass index (BMI) characterized as healthy (<25) or obese (≥30), and not clinically diagnosed as diabetic. All participants were Qatari nationals. Subjects who had a history of any metabolic disease or had consumed antibiotics or steroids in the preceding three months of sampling were excluded from this study. Seventy-three volunteers participated in this study, and were categorized according to their BMI into obese (*N* = 37) and lean control (N = 36) groups. Information on disease history, oral health (Appendix A), medication, feeding habits, physical activity, socioeconomic status, smoking habit, and family disease history was collected on the prescribed questionnaire. All participants in this study signed an informed consent form for the use of their information for sample analysis as anonymous volunteers. The institutional ethics review boards of the QBB (MOPH-QBB-IRB-011) and Qatar University (QU-IRB 969-A/18) approved this study in compliance with participant anonymity, research ethical, moral, and biosafety standards.

### 2.2. Plasma Biochemistry

Blood samples were collected in anticoagulant-coated evacuated tubes (BD, Mississauga, ON, Canada). Plasma concentrations of the hormones, enzymes, and lipid markers were analyzed at Hamad Medical Corporation (HMC) diagnostic laboratory using Cobas 6000 analyzer (Roche Diagnostics), as described previously [21,22,23]. A complete list of instruments and reagents used for plasma biochemistry is available in the Appendix A. 

### 2.3. 16S rRNA Sequencing

Saliva samples were collected from the participants by spitting saliva in sterile tubes. The samples were transported on ice from QBB to the Biomedical Research Center (BRC) of Qatar University (QU). Only 69 (Obese 36 and Control 33) saliva samples were available for sequencing as three participants did not provide a saliva sample, and we lost one sample during DNA extraction. Genomic DNA was extracted from the samples using a commercially available DNA extraction kit (QIAamp DNA Mini Kit, 51306, Germantown, MD, USA). The quality and quantity of the DNA were evaluated using NanoDrop-2000 (Thermo Fisher Scientific, Waltham Massachusetts, US) and Qubit-4 (Life Technologies, Carlsbad, California, US). The DNA samples were then subjected to 16S rRNA library preparation protocol using an Illumina Nextera XT Library Prep. Kit (FC-131-1002, Illumina Inc., San Diego, CA, USA). In brief, the V3–V4 region of the 16S rRNA gene was amplified using a 337F/805R primer pair [24], followed by an Illumina two-step amplification library preparation strategy [25]. Prepared libraries were cleaned and normalized using magnetic beads (Agencourt Ampure XP, Beckman Coulter, IN, USA). Finally, all libraries were pooled together in equal volumes and denatured using 0.2 N NaOH. The sequencing was performed on Illumina MiSeq (San Diego, CA, USA) using a 600 cycles v3 kit (MS-102-3003; Illumina, San Diego, CA, USA).

### 2.4. Bioinformatics

The data were obtained as paired-end reads. Forward and reverse reads were merged before analysis. The data were subjected to quality filtration and chimera removal using the DADA2 plugin implemented in QIIME2 [26,27]. The first thirteen bases of the forward and reverse reads were trimmed, while truncation was performed at 255 bases to allow sufficient overlapping of the forward and reverse reads. The DADA2 plugin generated 7110 sequence features, defined as unique 16S rRNA gene sequence variants. Phylogenetic diversity analysis was performed on QIIME2 using q2-phylogeny plugin that wraps mafft-fasttree program. Taxonomic classification was performed using Greengenes 13-8 database as the reference [28,29]. Each feature sequence was assigned taxonomy for >97% identity (or < 3% divergence) at the species, >95% at the genus, >90% at the family, >85% at the order, >80% at the class, and >77% at the phylum level [30]. 

### 2.5. Statistical Analysis

Demographic data were arranged from lowest to highest possible values and categorically numbered from 1 to 6 (Appendix A). The Mann–Whitney U test was performed to compare the mean differences in plasma biochemistry and demographic characteristics between the study groups. Spearman′s rank correlation coefficient was applied to measure the correlation between these variables. The Benjamini–Hochberg false discovery rate (FDR) was used to perform multiple comparisons of the *p*-values, and an adjusted *p* < 0.05 was considered statistically significant [31]. Core microbiome diversity analysis was performed at 28,153 sequencing depth using observed_OTUs, faith_PD, and Shannon indexes for alpha diversity analysis. Weighted_unifrac unweighted_unifrac, and Bray_Curtis_matrix indexes were used for beta diversity analysis to generate principal coordinates (PCoA) plots. The Wallis test was used to compare within-sample diversity (alpha diversity). Beta diversity (between samples) significance analysis was performed using the PERMANOVA test. For taxonomic assessment of microbiota, taxa were represented at a particular phylogenetic resolution (phylum, family, and genus). Only those taxa that had a relative abundance of at least 0.5% in any of the two groups were included in statistical analysis, and remaining data were discarded. The Kruskal–Wallis test was applied to compare taxonomic differences at all hierarchical levels. The association between microbiome and biochemical parameters was assessed using Spearman′s rank correlation coefficient and stepwise linear regression. A power calculation based on a previous similar study [32] indicated that a sample size of 35 per group has 95% power to detect a minimal difference of 11% in microbiome composition between diabetics and controls, with 10% deviation from the mean value (σ) at a level of α = 0.01 [33].

## 3. Results

The demographic and plasma biochemistry features of the study groups are presented in Table 1. The age distribution was similar across the two groups, with a mean age of 36.75 ± 8.0 years in the obese and 36.52 ± 7.9 years in the control group. The mean BMI was 35.65 ± 4.9 and 22.73 ± 1.5 for the obese and control groups, respectively. Fasting glucose and Hemoglobin A1c (HbA1c) levels were higher in the obese group compared to the control (*p* ≤ 0.001). Testosterone, sex hormone-binding globulin (SHBG), and insulin concentrations were lower (*p* ≤ 0.001), whereas estradiol hormone concentrations were higher (*p* = 0.03) in the obese group compared to the control group. No differences were observed for dietary habits, physical activity, smoking, and antibiotic usage between the study groups (Table 1). Spearman’s correlation analysis revealed a negative correlation between testosterone and BMI (r = −0.68, *p* ≤ 0.001) and insulin (r = −0.50, *p* ≤ 0.001). On the other hand, a positive correlation was observed between BMI and estradiol (r = 0.44, *p* ≤ 0.01) (Figure 1). 

High-throughput 16S rRNA amplicon sequencing yielded 6,790,910 sequences for all analyzed samples (*n* = 69, median ± SD = 96,330 ± 20,849.7). After quality control, 4,028,561 sequences of good quality (Phred quality score ≥ ASCII 30), belonging to 7110 features, were used for further analysis. All obtained sequences were demultiplexed and deposited to the NCBI Sequence Read Archive (SRA) for future reference under study accession number PRJNA587625. 

Figure 2 depicts the alpha diversity indexes of bacterial communities in the two study groups. The Faith_PD, Shannon, and observed_OTUs indexes were utilized to determine taxonomic diversity (species richness and evenness) within the samples. None of the diversity measures significantly differed between the obese and the control groups (Kruskal–Wallis *p* ≥ 0.05). Similarly, weighted_unifrac distance matrix-based PCoA plots showed that no distinct clustering pattern was present between the microbiome of the study samples (Figure 3). The Permutational multivariate analysis of variance (PERMANOVA) tests applied in beta diversity indexes did not show significant differences (PERMANOVA *p* = 0.37) among the studied groups.

The taxonomic analysis was performed using the greengenes 13-8 database as a reference. In total, 50 phyla, 124 classes, 188 orders, 412 families, 595 genera, and 676 species were identified in the 69 saliva samples. In descending order, the four most dominant phyla present in the oral microbiome of the study population were Firmicutes (43.3%), Bacteroidetes (25.2%), Proteobacteria (10.7%), and Actinobacteria (8.2%). *Streptococcus spp*., (22.2%) *Prevotella melaninogenica* (13.4%), and *Veillonella dispar* (6.6%) were the most abundant bacterial species that belonged to family *Streptococcaceae* (22.5%), *Prevotellaceae* (18.1%), and *Veillonellaceae* (10.1%), respectively. The Kruskal–Wallis test showed no differences (*p* ≥ 0.05) in microbiome population at any phylogenetic level between the studied groups (Figure 4). However, a generalized linear model revealed that the Firmicutes/Bacteroidetes ratio was significantly higher in obese IR subjects when compared to insulin sensitive control (2.25 ± 1.83 vs. 1.76 ± 0.58; corrected *p*-value = 0.04) after correcting for potential confounders including HbA1c, insulin, and triglycerides. Similarly, Fusobateria also exhibited a significant difference between the lean and obese groups (6.2 ± 4.3 vs. 4.5 ± 3.0, respectively; corrected *p*-value = 0.05) after correcting for potential confounders (HbA1c, insulin, and triglycerides) (Figure 5). 

Figure 6 presents Spearman’s correlation analysis between microbiome, population demographic characteristics, and plasma biochemistry data. Irrespective of the treatment group, certain bacterial phyla (Acidobacteria, Bacteroidetes, Fusobacteria, Proteobacteria, Spirochaetes, and Firmicutes/Bacteroidetes) were positively associated (corrected *p*-value = 0.05, r ≤ +0.5) with estradiol, fast food consumption, creatinine, breastfed during infancy, triglycerides, and thyroid-stimulating hormone (TSH) concentrations. Particularly, the phyla Proteobacteria and Acidobacteria were positively associated with elevated estradiol concentrations. The Firmicutes/Bacteroidetes ratio was negatively associated with HDL cholesterol and positively associated with TSH concentrations. In order to identify the best predictors of microbial taxon association with the demographic parameters, a stepwise linear regression was carried out. The regression model indicated that regardless of BMI, estradiol and HDL cholesterol were the best predictors of Acidobacteria, whereas estradiol, HDL cholesterol and triiodothyronine were the best predictors of Firmicutes. The model also revealed that creatinine was the best predictor of Fusobacteria regardless of BMI (Table 2). 

## 4. Discussion

To date, limited attention has been directed to studying the oral microbiome during the pre-diabetic phase, when metabolic and immune-inflammatory perturbation are silently underway. To fill this gap, we investigated the association between the oral microbiome and metabolic markers of clinically healthy adults categorized as obese hyperglycemic and healthy normoglycemic subjects in Qatar. 

We observed significant differences between the two investigated groups in terms of HbA1c, glucose, insulin, Homeostatic Model Assessment of Insulin Resistance (HOMA-IR), C-peptide, triglycerides, and HDL cholesterol levels, even though both the groups had no differences in demographic characteristics. On the other hand, we observed no association between these obesogenic parameters and daily activity, breastfed during infancy, smoking, and fast food consumption. Furthermore, we observed no association between BMI/hyperglycemia and thyroid gland activity, although this gland has a principal role in the regulation of cellular metabolism. It is worth noting that obese hyperglycemic males enrolled in this study had hypogonadism (drop in testosterone and SHBG concentrations) and elevated levels of the estradiol hormone. This observed negative association between testosterone and BMI/hyperglycemia is now a well-known phenomenon that describes a greater tendency to acquire hypogonadism in T2DM males [34]. Furthermore, a low testosterone concentration has been recognized as a reliable predictor of IR and the probability of developing T2DM in the future [35]. Meyer et al. (2017) suggested that in male obese subjects, testosterone is converted to estrogen by the aromatase enzyme produced in adipose tissue, which eventually results in male hypogonadism [36]. On the other hand, the estrogen hormone is a strong regulator of body weight and insulin sensitivity through the activation of G-protein-coupled estrogen receptors and the overactivation of SHBG [37]. Narrated together, all three sex hormones have a significant implication on T2DM development that corresponds directly to host metabolic rate and thyroid hormones activities [38]. Thyroid gland hormones are involved in cellular metabolism, and the overactivity of the gland is considered as a risk factor for diabetes development, especially in pre-diabetes subjects [39]. 

Since the microbiome has been enormously described as a co-regulator of host metabolism and adipogenesis [40], we ran microbiome analyses on saliva samples and studied its association with metabolic markers as indicated above. We observed no difference in microbiome alpha and beta diversity indexes between the two groups. Weighted_unifrac analysis of beta diversity revealed that approximately 60% of the samples were scattered along axis 1, which indicates that no changes in microbiome diversity as per hyperglycemia were present. However, after correcting for potential confounders, including HbA1c, insulin, and triglycerides, the Firmicutes/Bacteroidetes ratio was found to be significantly higher in obese IR subjects when compared to insulin-sensitive controls. Similar observations have previously been reported by Demmer et al. (2017), who studied the subgingival microbiome of oral infections, glucose intolerance and IR in non-diabetic adults [8]. The authors reported that the increase in Firmicutes or drop in Bacteroidetes population is associated with periodontitis and systemic inflammations [8]. An increase in the Firmicutes/Bacteroidetes ratio is considered a prognostic factor for the development of T2DM [41], as it is observed that most of the bacterial genera in the phylum Firmicutes can contribute to host weight gain and obesity [42].

In general, Firmicutes, Bacteroidetes, and Proteobacteria were the most abundantly present bacterial phyla in the salivary microbiome of our study samples. Previous studies also showed that these bacterial phyla are most abundantly present in our mucosal communities and are associated with host metabolic rate and energy homeostasis [42,43]. However, there are also contradictory findings in the literature about microbial phylogenetic association with T2DM. Xiao et al. (2017) found that the oral microbiome of diabetic and pre-diabetes mice groups is different from normoglycemic mice [44]. However, as we observed, they also found higher population abundances of Firmicutes, Bacteroidetes, and Proteobacteria. Similarly, Long et al. (2017) found that although Firmicutes are the most abundantly present bacteria in oral microbiome, bacterial taxa found in the phylum Actinobacteria are associated with the risk of T2DM development [18]. In contrast, Anbalagan et al. [43] observed that the oral microbiome of T2DM patients was not different from healthy controls. These controversies in observations could be attributed to different study designs and sampling methods. In our study, the lack of changes in microbiome alpha and beta diversity may be associated with the sampling method or demographic characteristics of the participants. Two major contributors to microbiome diversity, food and physical activity, were constant in our study population [45]. Furthermore, saliva samples were obtained using the spitting technique; however, many studies found that microbiome composition could change due to different sampling procedures: saliva/spitting, dental surface, inner cheeks, and lingual swabbing were different [46,47]. 

We performed a correlation analysis between demographic data and the microbiome at the phylum level. It was observed that the phylum Firmicutes was negatively associated with estradiol hormone levels. The Firmicutes/Bacteroidetes ratio was positively associated with triglycerides and TSH concentrations and negatively associated with HDL cholesterol. However, no correlation between microbiome diversity and BMI, plasma glucose, or Hb1Ac was observed. Therefore, to observe the predictors of microbiome change, the subsequent analysis of microbiome association was performed without consideration of confounding factors. The regression model indicated that regardless of BMI, estradiol and HDL cholesterol were the best predictors of Acidobacteria, whereas estradiol, HDL cholesterol, and triiodothyronine were the best predictors of Firmicutes. The model also revealed that creatinine was the best predictor of Fusobacteria regardless of BMI. These phyla are associated with host metabolism and energy harvest and, therefore, may support obesogenic phenotype and IR [48]. Previously, similar observations were reported by Si et al. (2017), who researched the oral microbiome when exploring biomarkers of metabolic syndrome and reported a linear correlation of HDL cholesterol and triglycerides with members of the phyla Firmicutes and Proteobacteria [49]. 

In conclusion, we observed that obese pre-diabetic male subjects had significantly low testosterone and sex hormone-binding globulin that may compromise their sexual activity. Overall, oral microbial ecology was highly diverse, including 7110 features at different hierarchal levels. Certain bacterial phyla were associated with reproductive and metabolic hormones, triglycerides, and HDL cholesterol concentrations. Significant differences in the Firmicutes/Bacteroidetes ratio were observed between the pre-diabetic and control groups. The association between the oral microbiome and host metabolic health identified in this study may be advantageous as the key aid in the early prediction of T2DM. More studies with larger samples are warranted to confirm these findings in both genders and different ethnicities.

## Figures and Tables

**Figure 1 microorganisms-07-00645-f001:**
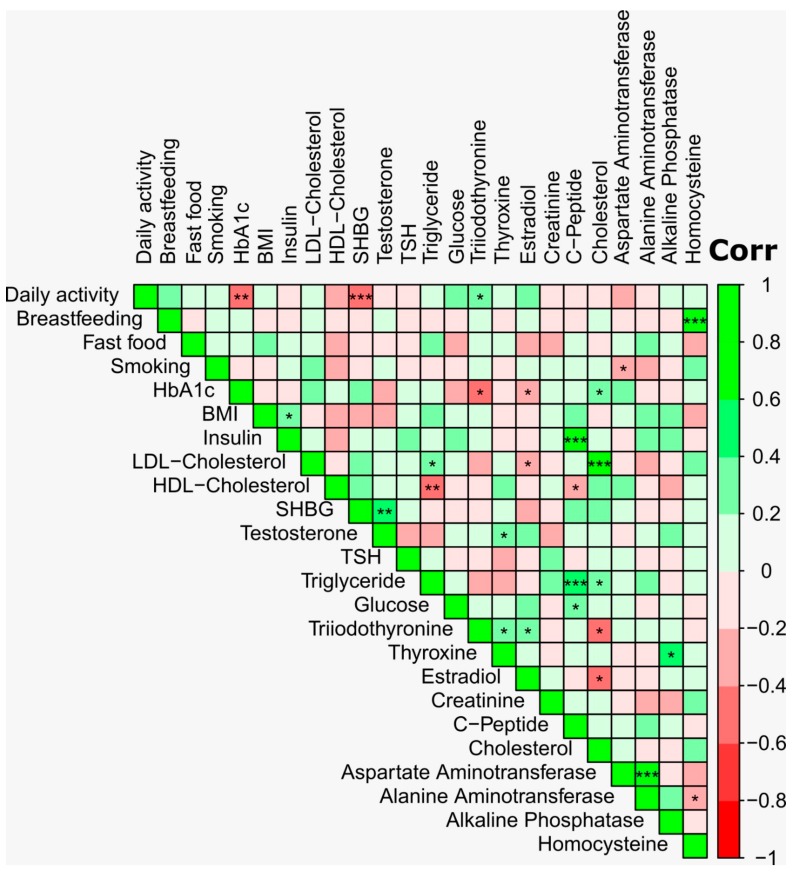
Spearman’s correlation analysis was applied for the pairwise analysis of plasma biochemical profile and demographic characteristics. The Benjamini–Hochberg false discovery rate (FDR) was used to perform multiple comparisons of the *p*-values. The color intensity shows the strength of correlation. Asterisks in each box indicate the corrected *p*-value; *** ≤ 0.001, ** ≤ 0.01 and * ≤ 0.05. Analyses were performed using the corrplot package in RStudio version 3.5.0.

**Figure 2 microorganisms-07-00645-f002:**
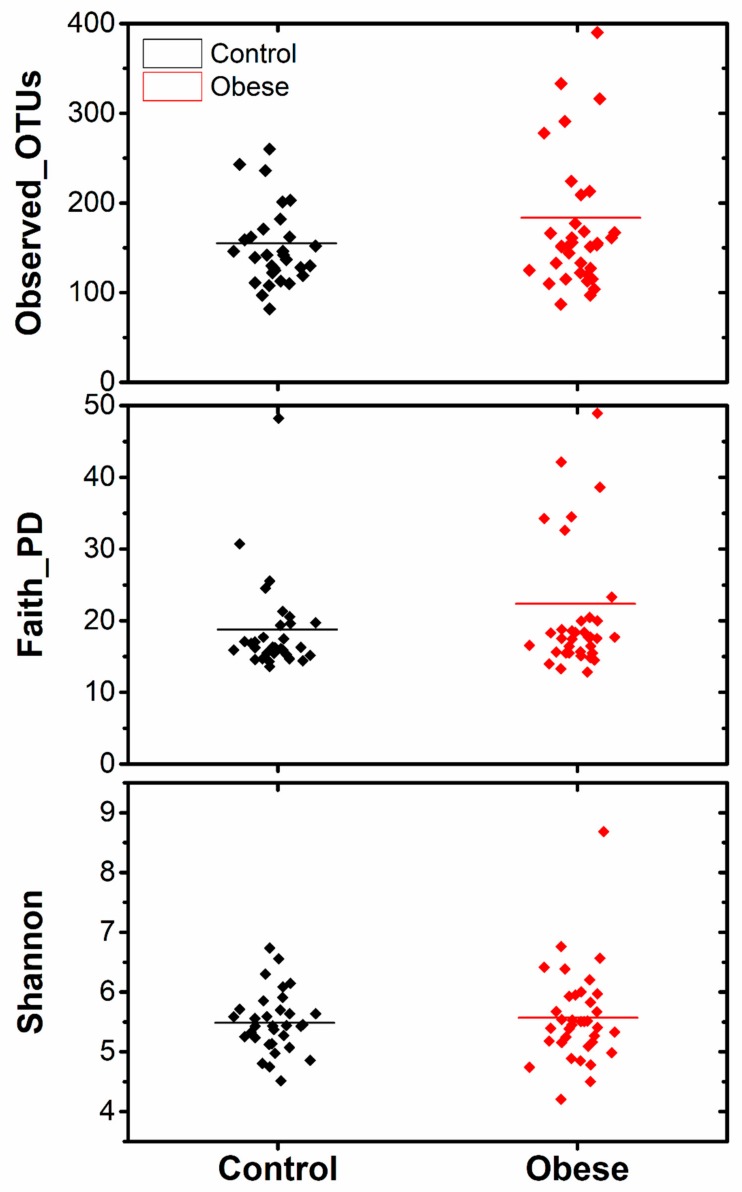
Alpha diversity analysis at 28,153 sequencing depth for observed_OTUs, faith_PD, and Shannon indexes.

**Figure 3 microorganisms-07-00645-f003:**
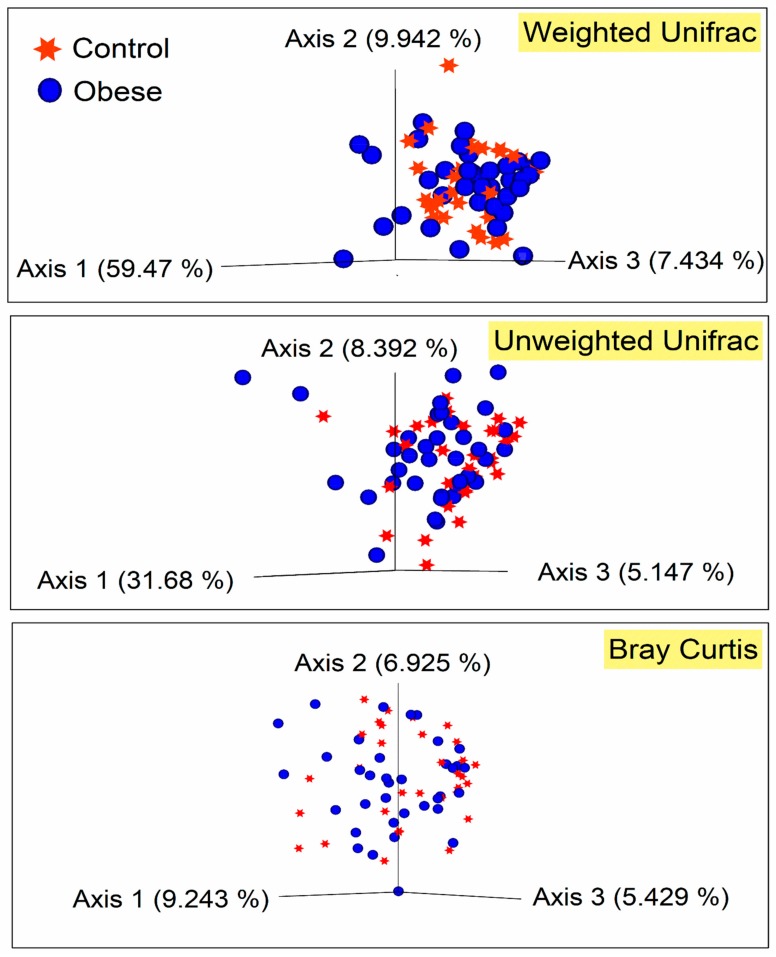
Beta diversity analysis on a 3D principal coordinates (PCoA) plot for weighted_unifrac, unweighted_unifrac and Bray Curtis indexes.

**Figure 4 microorganisms-07-00645-f004:**
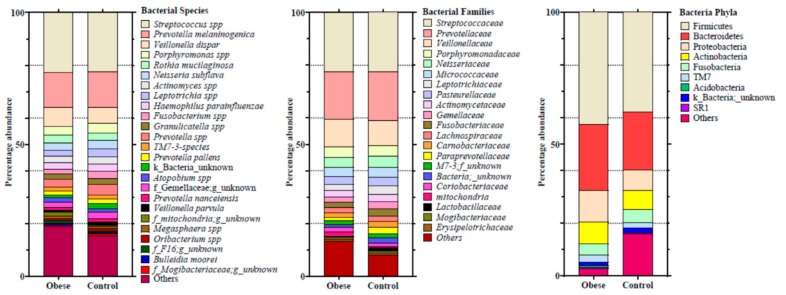
Taxonomic distribution of the microbiome at the phylum, family and species level. Height of each bar represents the relative proportion of that phylotype. Only microbes with a relative proportion of at least 0.5% are presented in the bar plots. The Kruskal–Wallis test shows no differences (*p* ≤ 0.05) between the study groups.

**Figure 5 microorganisms-07-00645-f005:**
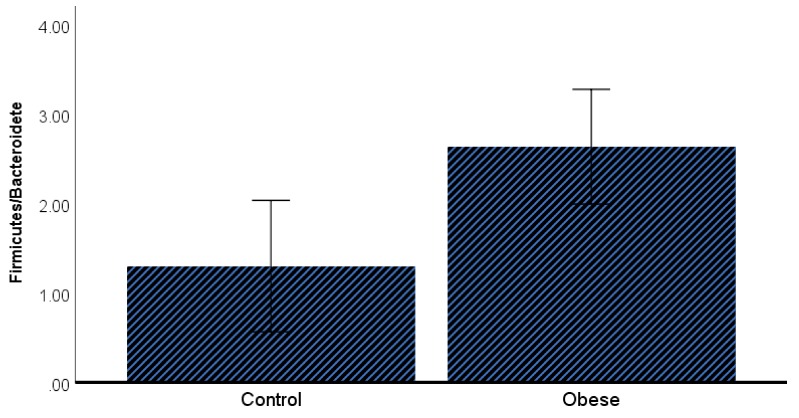
A bar chart of the estimated means of the Firmicutes/Bacteroidetes ratio and Fusobacteria in the control and obese groups after adjusting for triglycerides, insulin and HBA1c. Data are presented as the mean (95%CI).

**Figure 6 microorganisms-07-00645-f006:**
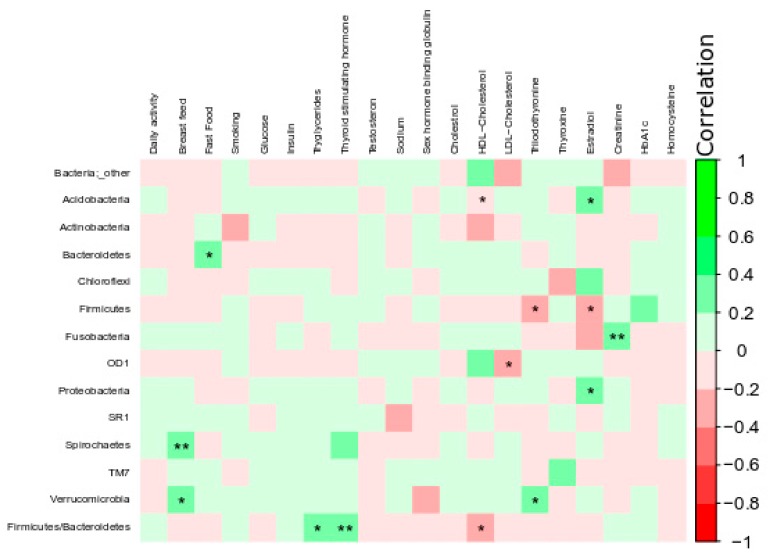
Spearman’s correlation analysis between bacterial phylum and demographic/plasma biochemistry values. The color intensity indicates the strength of the correlation depicted as r-value. Analyses were performed using the corrplot package in RStudio version 3.5.0. Asterisks in each box indicate the corrected *p*-value; ** ≤ 0.01 and * ≤ 0.05.

**Table 1 microorganisms-07-00645-t001:** Demographic characteristics * and plasma biochemistry profile of the study cohort.

Study Parameter	Study Group		Corrected *p*-Value
	Obese	Control	
BMI	35.65 ± 4.92	22.73 ± 1.52	< 0.001
Hemoglobin A1c (HbA1c) (%)	5.68 ± 0.51	5.14 ± 0.76	< 0.001
Fasting glucose (mmol/L)	5.96 ± 0.42	4.80 ± 0.35	< 0.001
Insulin (umol/L)	16.56 ± 7.95	6.42 ± 1.86	< 0.001
Homeostatic Model Assessment of Insulin Resistance (HOMA-IR)	4.3 ± 1.7	1.3 ± 0.3	< 0.001
Triglyceride (mmol/L)	1.74 ± 1.12	1.18 ± 0.74	0.051
Total cholesterol (mmol/L)	5.09 ± 0.97	5.32 ± 1.03	0.382
HDL cholesterol (mmol/L)	1.11 ± 0.28	1.32 ± 0.43	0.772
LDL cholesterol (mmol/L)	3.23 ± 0.89	3.47 ± 0.86	0.215
Thyroid-stimulating hormone (mIU/L)	1.91 ± 1.38	1.95 ± 1.48	0.890
Triiodothyronine (pmol/L)	4.21 ± 0.70	3.99 ± 0.64	0.213
Thyroxine (pmol/L)	12.62 ± 1.41	13.01 ± 1.48	0.208
Testosterone (nmol/L)	13.38 ± 3.67	22.74 ± 7.35	0.005
Sex hormone-binding globulin (nmol/L)	26.20 ± 12.01	37.00 ± 16.28	0.034
Estradiol (pmol/L)	111.43 ± 42.26	92.27 ± 36.89	0.030
Creatinine (µmol/L)	72.05 ± 8.72	77.53 ± 9.19	0.027
C-peptide (ng/mL)	3.04 ± 0.96	1.51 ± 0.50	0.048
Aspartate aminotransferase (U/L)	24.78 ± 9.84	21.61 ± 8.74	0.120
Alanine aminotransferase (U/L)	39.68 ± 18.97	24.75 ± 13.86	0.004
Alkaline phosphatase (U/L)	73.92 ± 14.35	68.36 ± 12.83	0.111
Homocysteine (umol/L)	9.65 ± 2.54	10.53 ± 2.85	0.216
* Daily activity	2.39 ± 1.61	3.00 ± 1.66	0.195
* Breastfed in infancy	1.12 ± 0.33	1.00 ± 0.00	0.137
* Fast food consumption	2.53 ± 1.42	3.22 ± 1.55	0.212
* Smoking	1.19 ± 1.24	1.51 ± 1.36	0.343
* Antibiotic usage in last one year	2.39 ± 1.04	1.49 ± 0.50	0.204

Data are the mean ± standard deviation. The Mann–Whitney U test was performed to compare mean differences between the groups. The Benjamini–Hochberg false discovery rate (FDR) was used to perform multiple comparisons of the *p*-values. * Data provenance: The demographic data were collected as per the questionnaire designed by Qatar Biobank. For statistical analysis, the data were categorically arranged from 1 to 6, where 1 is lowest or negative value and 6 is the highest possible value. Please refer to the Appendix A for data provenance.

**Table 2 microorganisms-07-00645-t002:** Predictors of microbial taxa by stepwise linear regression after correcting for all potential confounders (BMI, lipids, glucose, and insulin).

Bacterium	Predictor	Adjusted R Square	Std. Error of the Estimate	*p*-Value
**Acidobacteria**	Estradiol	0.067	1.7	0.023
	Estradiol	0.13	1.7	0.005
	HDL cholesterol			0.024
**Firmicutes**	Estradiol	0.097	9.7	0.008
	Estradiol	0.141	9.4	0.002
	HDL cholesterol			0.046
	Estradiol	0.186	9.2	0.003
	HDL cholesterol			0.028
	Triiodothyronine			0.043
**Fusobacteria**	Creatinine	0.103	3.6	0.006

A stepwise linear regression analysis was performed on the bacterial taxa that showed significant association with different plasma biochemistry markers.

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
