# Peer review of "Profiling the Oral Microbiome and Plasma Biochemistry of Obese Hyperglycemic Subjects in Qatar"

_microorganisms, 2019, doi:10.3390/microorganisms7120645_

Round 1
Reviewer 1 Report
In this manuscript by Sohail et al., demographic measurements, plasma biochemistry, and composition differences in the oral microbiome were examined in the context of obesity and pre-diabetes. The goal of the research was to identify whether there are microbiome or other markers of the pre-diabetic state and/or obesity. While several associations were identified with serum metabolites and hormones, significant microbiome associations were not identified. The study asks an important and potentially actionable question, and the dataset they collected is quite useful for answering those questions. That being said, I have several suggestions for ways to improve the statistical rigor of this study and potentially uncover associations not reported currently:
Major Comments:
Throughout manuscript, correction for multiple testing needs to be applied to p-values. This should be done for analysis of demographic characteristics, serum profiling, and microbiome profiling. For Pearson correlations, one assumption is that variables need to be normally distributed. Please confirm that all variables were normally distributed or normalized/transformed prior to performing correlation test. If not, then a non-parametric Spearman correlation should be used. I would suggest applying a different model to test for microbiome associations with either obesity or pre-diabetes. Given that the microbiome was associated with several serum factors (and assuming those associations remain significant after correcting for multiple testing), a more powerful approach would be to control for those factors in a model. For example, test to see whether a microbial taxon is associated for obesity, while including HBA1C, insulin level, and Triglycerides in the model. This could be done either using a linear or generalized linear model or by using software like DEseq2. It’s possible that since the sample size is a bit on the low end, that some of these factors are contributing enough noise that it’s masking potential associations. Additionally, given that the sample size is a bit small, it’s possible that the study is just a bit underpowered to detect significant differences. I would suggest performing an analysis to identify what sample size would be needed to identify significant differences between groups given the current effect sizes.
Minor Comments:
Grammatical errors/awkward wording exists throughout the manuscript. Please correct. It is stated that oral and gut microbiomes can provide similar information (lines 60-61). Additionally, oral results from this study are compared with gut results from previous work (lines 260 – 281). These assertions are problematic. The oral and gut microbiota are quite distinct and don’t necessarily mirror each other. If there is some documented predictive value in comparing the two sites, please refer to those studies in the introduction. Otherwise, it is not useful to compare and contrast specific phylum-level changes between the two body sites. I would remove all reference to gut microbiomes when discussing the oral microbiome results (other than perhaps to state that there are differences in gut microbiomes in obesity/pre-diabetes). The final paragraph of the introduction includes the following two sentences: “However, it is unclear whether oral microbiome is per se associated with T2DM pathogenesis or the consequence of diabetes. To address this gap in knowledge we studied…” This makes it sound like the current study is addressing correlation vs. causation (which it cannot). I would reword this section to reflect that this study is attempting to identify oral microbiome markers of obesity and diabetes. Supplementary file 2 (referred to on line 93) was not included in submission. Please update the term “metagenomics” used throughout the manuscript to “16S rRNA gene sequencing” or “microbiome profiling.” “Metagenomics” is the study of all DNA/genomes from a sample (aka shotgun sequencing, not amplicon sequencing). See the article by Marchese and Ravel for precise definitions (DOI: 10.1186/s40168-015-0094-5). The filtering threshold as described on lines 132 – 134 is unclear as written. Is the minimum relative abundance of 0.5% in any one sample? An average across all samples within each group? Total across all samples within each group? Line 140: Fill in threshold (P<value) Table 1 (and throughout text): P-values can’t be zero. Please replace zero with < [minimum p-value could be] Figure 1: Multiple levels of significance are indicated in the plot, but only (*) is included in the legend. Please include a definition for all star levels in legend. Figure 3: I suggest including 2D PCoA plots rather than 3D PCoA plots. It is difficult for a reader to interpret 3D plots printed on a 2D page. Please include SRA accession number (currently PRJXXXXX)
Author Response
Comments and Suggestions for Authors
In this manuscript by Sohail et al., demographic measurements, plasma biochemistry, and composition differences in the oral microbiome were examined in the context of obesity and pre-diabetes. The goal of the research was to identify whether there are microbiome or other markers of the pre-diabetic state and/or obesity. While several associations were identified with serum metabolites and hormones, significant microbiome associations were not identified. The study asks an important and potentially actionable question, and the dataset they collected is quite useful for answering those questions. That being said, I have several suggestions for ways to improve the statistical rigor of this study and potentially uncover associations not reported currently:
Major Comments:
Throughout manuscript, correction for multiple testing needs to be applied to p-values. This should be done for analysis of demographic characteristics, serum profiling, and microbiome profiling.
Authors Response: We have now performed correction for multiple testing using Benjamini-Hochberg’s false discovery rate for analysis of demographic characteristics, serum profiling, and microbiome profiling as shown in Table 1, Figure 1 and Figure 5.
For Pearson correlations, one assumption is that variables need to be normally distributed. Please confirm that all variables were normally distributed or normalized/transformed prior to performing correlation test. If not, then a non-parametric Spearman correlation should be used.
Authors Response: We have now applied the non-parametric Mann-Whitney U test as well as Spearman’s correlation for all skewed data from serum biochemistry, demographic and microbiome measurements.
I would suggest applying a different model to test for microbiome associations with either obesity or pre-diabetes. Given that the microbiome was associated with several serum factors (and assuming those associations remain significant after correcting for multiple testing), a more powerful approach would be to control for those factors in a model. For example, test to see whether a microbial taxon is associated for obesity, while including HBA1C, insulin level, and Triglycerides in the model. This could be done either using a linear or generalized linear model or by using software like DEseq2. It’s possible that since the sample size is a bit on the low end, that some of these factors are contributing enough noise that it’s masking potential associations. Additionally, given that the sample size is a bit small, it’s possible that the study is just a bit underpowered to detect significant differences. I would suggest performing an analysis to identify what sample size would be needed to identify significant differences between groups given the current effect sizes.
Authors Response:
We would like to thank the reviewer for raising this important point. As requested, we conducted a generalized linear model to identify differences in microbiome between the two studied groups after correcting for HBA1C, insulin level, and Triglycerides. The model revealed that Firmicutes/Bacteroidetes ratio was significantly higher in the obese insulin resistant group when compared to insulin sensitive controls (2.25±1.83 vs 1.76±0.58, p=0.008). We have now added the new data in abstract, results and discussion as highlighted. Lastly, as suggested by the reviewer, a power calculationas shown previously [40]. Power calculation indicated that sample size of 35 per group has 95% power to detect a minimal difference of 11% in mean bacteria content between diabetics and controls with 10% deviation from mean value (σ) at a level of α = 0.01.
Minor Comments:
Grammatical errors/awkward wording exists throughout the manuscript. Please correct.
Authors Response: Authors made further efforts to improve manuscript English and removed awkward words.
It is stated that oral and gut microbiomes can provide similar information (lines 60-61). Additionally, oral results from this study are compared with gut results from previous work (lines 260 – 281). These assertions are problematic. The oral and gut microbiota are quite distinct and don’t necessarily mirror each other. If there is some documented predictive value in comparing the two sites, please refer to those studies in the introduction. Otherwise, it is not useful to compare and contrast specific phylum-level changes between the two body sites. I would remove all reference to gut microbiomes when discussing the oral microbiome results (other than perhaps to state that there are differences in gut microbiomes in obesity/pre-diabetes).
Authors Response: Zhang et al., 2015 (The oral and gut microbiomes are perturbed in rheumatoid arthritis and partly normalized after treatment." Nature medicine) suggest that a concordance was observed between the gut and oral microbiomes of the studied population, suggesting overlap in the abundance and function of species at different body sites. Furthermore, in this study authors reported dysbiosis of the gut and oral microbiomes in rheumatoid arthritis patients. However, as per your suggestion, authors removed all references from the discussion that compared this study’s findings with the gut microbiome results of the other studies.
The final paragraph of the introduction includes the following two sentences: “However, it is unclear whether oral microbiome is per se associated with T2DM pathogenesis or the consequence of diabetes. To address this gap in knowledge we studied…” This makes it sound like the current study is addressing correlation vs. causation (which it cannot). I would reword this section to reflect that this study is attempting to identify oral microbiome markers of obesity and diabetes.
Authors repose: We attempted to transform the study hypothesis as to saying that the study attempted to identify oral microbiome markers of obesity and diabetes.
Supplementary file 2 (referred to on line 93) was not included in submission.
Authors Response: We included references from the previous study for these analysis and have also included list of instruments used for these analysis in the supplementary file.
Please update the term “metagenomics” used throughout the manuscript to “16S rRNA gene sequencing” or “microbiome profiling.” “Metagenomics” is the study of all DNA/genomes from a sample (aka shotgun sequencing, not amplicon sequencing). See the article by Marchese and Ravel for precise definitions (DOI: 10.1186/s40168-015-0094-5).
Authors Response: In the revised manuscript we have replaced term “Metagenomics” with the microbiome.
The filtering threshold as described on lines 132 – 134 is unclear as written. Is the minimum relative abundance of 0.5% in any one sample? An average across all samples within each group? Total across all samples within each group?
Authors Response: Average 0.5% across all samples within each group was taken as threshold value for diagrammatic presentation and statistical analysis. As Stated in the results section, in total, 50 phyla, 124 classes, 188 orders, 412 families, 595 genera, and 676 species. It was not possible to describe them all in a figure or table.
Line 140: Fill in threshold (P<value) Table 1 (and throughout text): P-values can’t be zero. Please replace zero with < [minimum p-value could be]
Authors Response: We have revised statistical analysis and also we have revised p values.
Figure 1: Multiple levels of significance are indicated in the plot, but only (*) is included in the legend. Please include a definition for all star levels in legend.
Authors Response: We have revised figure and accordingly described them in figure legends
Figure 3: I suggest including 2D PCoA plots rather than 3D PCoA plots. It is difficult for a reader to interpret 3D plots printed on a 2D page.
QIIME2, the program that we used to generate PCoA plot uses Emperor software for PCoA plot. So we are unable to modify this graph due to QIIME2 limitations.
Please include SRA accession number (currently PRJXXXXX)
We have uploaded sequences to SRA and put project ID in the MS.
Reviewer 2 Report
My primary issue with this study is the final conclusion in Lines 286-289. Considering this study was conducted only at a single location on only males of a single ethnic community and only directed at the pre-diabetic group, these lines can be very misleading and should be reworded keeping in mind these limiting factors. Line 110: The use of the word metagenomics is incorrect here, since this refers to only the data analysis of amplicon sequencing data. Please correct this. Lines 94-109: There is no mention of any controls here. It needs to be clarified if adequate controls were in place for the amplicon sequencing Line 140: an actual value needs to be provided to compare the p-value Table 1 refers to "breastfeeding" in the demographic characteristics. This is very confusing to the reader since all the subjects were adult males. Assuming this refers to whether they were breastfed as infants, please reword this appropriately. Line 158: This also part of the results. Why does this have a separate heading? Line 163: A valid accession number needs to be provided here. Many references are incomplete and need to be corrected. A number of language changes need to be made across the manuscript:
- There are articles missing in sentences throughout the manuscript (including the very first line of the abstract).Please review the English language thoroughly.
- Line 42 needs rephrasing
Line 52: ...approaches that "use"....
Line 54: ...s strong correlation...
Line 58-61: Rewrite
Lines 90-91: Rewrite
Line 92: remove the word "against"?
Lines 212 & 216: replace "despite that" with "even though"
Line 216: "principal"
Lines 230-232: This should be one sentence
Line 235: please reword
Line 269: Previous studies "report"...
Author Response
Comments and Suggestions for Authors
My primary issue with this study is the final conclusion in Lines 286-289. Considering this study was conducted only at a single location on only males of a single ethnic community and only directed at the pre-diabetic group, these lines can be very misleading and should be reworded keeping in mind these limiting factors.
We have changed this conclusion in the abstract and in conclusion by adding a statement that more studies are warranted to confirm these findings using both genders and different ethnicities.
Line 110: The use of the word metagenomics is incorrect here, since this refers to only the data analysis of amplicon sequencing data. Please correct this.
Authors Response: We have removed word metagenomics from the entire draft. The word metagenomics is replaced by microbiome.
Lines 94-109: There is no mention of any controls here. It needs to be clarified if adequate controls were in place for the amplicon sequencing.
Authors Response: We have added information about Obese (36) and control (33) subjects
Line 140: an actual value needs to be provided to compare the p-value
Authors Response: We have corrected this value.
Table 1 refers to "breastfeeding" in the demographic characteristics. This is very confusing to the reader since all the subjects were adult males. Assuming this refers to whether they were breastfed as infants, please reword this appropriately.
Authors Response: As suggested by the reviewer, we reworded this information as breastfed and explained
Line 158: This also part of the results. Why does this have a separate heading?
Authors Response: We removed this heading as per reviewer suggestion.
Line 163: A valid accession number needs to be provided here.
Authors Response: We are working o NCBI submission.
Many references are incomplete and need to be corrected.
Authors Response: References has been corrected after removing file from the endnote.
A number of language changes need to be made across the manuscript:
There are articles missing in sentences throughout the manuscript (including the very first line of the abstract).Please review the English language thoroughly.
Line 42 needs rephrasing.
Authors Response: Sentence has been rephrased.
Line 52: ...approaches that "use"....
Authors Response: Correction has been made
Line 54: ...s strong correlation...
Authors Response: Correction has been made
Line 58-61: Rewrite
Authors Response: Sentence has been rephrased.
Lines 90-91: Rewrite
Authors Response: Sentence has been rephrased.
Line 92: remove the word "against"?
Authors Response: Word has been removed.
Lines 212 & 216: replace "despite that" with "even though"
Authors Response: Change has been made
Line 216: "principal"
Authors Response: Correction has been made
Lines 230-232: This should be one sentence
Authors Response: Both lines are combined and rephrased.
Line 235: please reword
Authors Response: Sentence has been rephrased
Line 269: Previous studies "report"...
Authors Response: Correction made
Reviewer 3 Report
General comments:
The authors compared oral microbiome in 37 obese and 36 lean males, and compared the bacteria taxa with the biochemical variables of the participants in a cross section study. No significant difference in oral microbiota was detected between the two groups. Obese subjects had higher testosterone and sex hormone binding globulin (SHBG) concentrations compared to the control group. Negative association was dftected between BMI and testosterone and SHBG. Proteobacteria and Acidobacteria were positively associated with elevated Estradiol concentrations and Firmicutes/Bacteroidetes were associated with triglyceride and thyroid hormones concentrations. Limited novel information was reported in this study.
Specific comments:
The authors indicated data from 4 participants have not been included. The numbers of obese and lean participants whose data were included in the final analysis should be indicated in the text or Table 1. No female was included in the study is a limitation, which will restrict the application of the findings from this study. Oral hygiene may substantially affect the abundance of bacterial in slavery. It will be ideal to include that information. Oral microbiota may be directly influence by slavery glucose. The history of medication, including recent administration of antibiotics, has not been described. Minors:Fig 5, vertical legend -0.3- -0.8 were merged with ticks. Table 1: Breastfeeding, may be better replaced by breastfed Fonts difference in Line 216.
Author Response
Comments and Suggestions for Authors
General comments:
The authors compared oral microbiome in 37 obese and 36 lean males, and compared the bacteria taxa with the biochemical variables of the participants in a cross section study. No significant difference in oral microbiota was detected between the two groups. Obese subjects had higher testosterone and sex hormone binding globulin (SHBG) concentrations compared to the control group. Negative association was dftected between BMI and testosterone and SHBG. Proteobacteria and Acidobacteria were positively associated with elevated Estradiol concentrations and Firmicutes/Bacteroidetes were associated with triglyceride and thyroid hormones concentrations. Limited novel information was reported in this study.
Authors response: Thanks for reviewing our manuscript. While we agree with most of the reviewers’ comments, it is important to mention that to the best of our knowledge this is the very first study of human microbiome from Qatari population. Furthermore very few studies have reported oral as well as gut microbiome of pre-diabetes/hyperglycemia stage (Allin et al., 2018, Li-Fang Yeo et al., 2019) and compared them extensively with the demographic characteristics.
Specific comments:
The authors indicated data from 4 participants have not been included. The numbers of obese and lean participants whose data were included in the final analysis should be indicated in the text or Table 1.
Authors response: We have added this information in the main text under “16S rRNA Sequencing” heading.
No female was included in the study is a limitation, which will restrict the application of the findings from this study.
Authors Response: We appreciate your comments and are planning to extend our work while including both genders. We also revised our conclusions accordingly.
Oral hygiene may substantially affect the abundance of bacterial in slavery. It will be ideal to include that information. Oral microbiota may be directly influence by slavery glucose.
We collected information on oral health of the participants. Now we have added this information in the results section.
The history of medication, including recent administration of antibiotics, has not been described.
Information on antibiotic usage in the past one year was collected and is now reported in table 1. However, only participants with no antibiotic usage in the past three month were enrolled in the study.
Minors:Fig 5, vertical legend -0.3- -0.8 were merged with ticks.
Authors Response: We have corrected formatting of the graph.
Table 1: Breastfeeding, may be better replaced by breastfed
Authors Response: Correction made.
Fonts difference in Line 216.
Authors Response: correction made.
Round 2
Reviewer 1 Report
The authors did an excellent job addressing the comments I raised in the initial submission of this manuscript. Most of the concerns I had have been thoroughly addressed, including the use of non-parametric statistics, adding in a GLM with relevant covariates, as well as a number of the minor comments.
Several technical issues are still outstanding (or have been introduced), however. They should be relatively straight forward to address:
Lines 151 – 152: Correct “P >= 0.01” to “P <= 0.01”. Additionally, it would be helpful to write something like “corrected P” anywhere a corrected P value is referenced. Legend of Figures 1 and 2: The definitions of the stars are now included, but it’s a bit confusing as written, Please include the “P <= “ (right now it follows right after discussion about r, so it looks like r <= 0.001”. Also, the final category should be P <= 0.05 (rather than P > 0.05). Are the new P-values reported from the GLM corrected for multiple testing? They should be, and if so, please include that in the text/table 2.
Author Response
Reviewer 1:
Lines 151 – 152: Correct “P >= 0.01” to “P <= 0.01”.
Authors response: Corrections have been made and track changed
Additionally, it would be helpful to write something like “corrected P” anywhere a corrected P value is referenced.
Authors response: Corrected P-values have been indicated in figure legends and in text in the results section. Track changed
Legend of Figures 1 and 2: The definitions of the stars are now included, but it’s a bit confusing as written, Please include the “P <= “ (right now it follows right after discussion about r, so it looks like r <= 0.001”. Also, the final category should be P <= 0.05 (rather than P > 0.05).
Authors response: Statement has been rephrased as “Asterisks in each box indicate corrected P-value; *** ≤ 0.001, ** ≤ 0.01 and * ≤ 0.05.” Track changes in figure legends.
Are the new P-values reported from the GLM corrected for multiple testing? They should be, and if so, please include that in the text/table 2.
Authors response: P values for GLM are corrected and indicated in the text. Track changed
Reviewer 2 Report
I am happy with the revised version submitted by the authors.
Author Response
Respected reviewer we appreciate your time and efforts for improving manuscript.
Reviewer 3 Report
I think this manuscript can be accepted for publication.
Author Response
Respected reviewer we appreciate your time and efforts for improving our manuscript.